# Amphipods in Mediterranean Marine and Anchialine Caves: New Data and Overview of Existing Knowledge

Carlos Navarro-Barranco [1,*], Alejandro Martínez [2], Juan Sempere-Valverde [1], Sahar Chebaane [3,4], Markos Digenis [5], Wanda Plaitis [6], Eleni Voultsiadou [7] and Vasilis Gerovasileiou [5,6]

[1] Laboratorio de Biología Marina, Facultad de Biología, Universidad de Sevilla, Avenida Reina Mercedes, 41012 Sevilla, Spain; juansempere91@gmail.com

[2] Molecular Ecology Group (MEG), Water Research Institute (IRSA), National Research Council of Italy (CNR), 28922 Verbania Pallanza, Italy; amartinez.ull@gmail.com

[3] Marine and Environmental Sciences Centre (MARE), ARNET—Aquatic Research Network, Regional Agency for the Development of Research, Technology and Innovation (ARDITI), Edifício Madeira Tecnopolo, Caminho da Penteada, 9020-105 Funchal, Portugal; sahar1994ch@gmail.com

[4] Departamento de Biologia Animal, Faculdade de Ciências, Universidade de Lisboa, Campo Grande, 1749-016 Lisboa, Portugal

[5] Department of Environment, Faculty of Environment, Ionian University, 29100 Zakynthos, Greece; markosdigenis@gmail.com (M.D.); vgerovas@ionio.gr (V.G.)

[6] Hellenic Centre of Marine Research (HCMR), Institute of Marine Biology, Biotechnology and Aquaculture (IMBBC), 71500 Heraklion, Greece; wanda@hcmr.gr

[7] Department of Zoology, School of Biology, Aristotle University of Thessaloniki, 54124 Thessaloniki, Greece; elvoults@bio.auth.gr

* Correspondence: carlosnavarro@us.es

**Abstract:** Marine and anchialine caves host specialized faunal communities with a variable degree of endemism and functional specialization. However, biodiversity assessments on this habitat are scarce, particularly in relation to small-sized cryptic fauna (such as amphipods), which often play a key role in benthic ecosystems. The present article compiles all records of marine and brackish-water amphipods inhabiting marine and anchialine caves along the Mediterranean basin, combining information extracted from a literature review with newly acquired records. A total of 106 amphipod species has been reported (representing approximately 20% of the Mediterranean amphipod species), mostly from the North-Western Mediterranean. Examination of new material from marine caves in Greece has yielded 14 new records from the East Ionian and Aegean Sea. Most of the reported species display wide ecological amplitude in terms of habitat and substrate preferences, feeding habits as well as bathymetric and geographical distribution. In contrast, only 17 amphipod species have been reported from marine-brackish waters in anchialine caves, predominantly represented by cave specialists with a narrow spatial distribution and distinct morphological traits. Our overall knowledge on amphipods inhabiting Mediterranean caves is far from complete so that new and valuable findings are expected to occur as new caves are explored.

**Keywords:** Crustacea; biodiversity; benthic ecology; stygobionts; biogeography; Mediterranean Sea

## 1. Introduction

Motile macrofauna constitute a ubiquitous and key component of all coastal marine ecosystems. Many motile invertebrate species live in association with living primary substrates (e.g., macroalgae, seagrasses, and sessile invertebrates), whereas others are specialised to burrow or crawl amongst seafloor sediments or the small crevicular spaces within rocky or biogenic reefs [1]. Those epifaunal and infaunal communities frequently account for the greatest proportion of invertebrate species and abundance in benthic communities and, given their rapid temporal turnover rates, they are major contributors to secondary production in a wide range of habitats and geographical regions [1–6].

Amphipod crustaceans dominate benthic macroinvertebrate communities both in terms of abundance and numbers of species. The Order Amphipoda comprises approximately ten thousand species described within more than 200 families [7]. They inhabit both marine, brackish and freshwater habitats at all latitudes and bathymetric ranges, although the highest species richness occurs in benthic marine shallow water environments [7,8]. The numerical dominance of amphipods in comparison with other major macrofaunal groups (e.g., molluscs, annelids and other crustaceans) has been reported in a wide variety of habitats such as macroalgal and seagrass beds, gorgonian forests, fouling communities, as well as intertidal and shallow unvegetated sediments [5,9–14]. Many amphipod mesograzers affect macrophyte biomass and, as a prey source in benthic ecosystems, they constitute a relevant link between primary producers and higher-order consumers [15–17]. On the other hand, suspension-feeding amphipods also contribute to benthic–pelagic coupling and carbon dynamics by consuming a significant quantity of particles from the water column [18,19]. Amphipod species also exhibit a great diversity of life history strategies and variability in their tolerance to environmental conditions [20,21]. For all these reasons, they are reliable bioindicators used in laboratory ecotoxicology tests, as well as in field biomonitoring studies [21]. Unfortunately, amphipod communities are often overlooked in biodiversity inventories and ecological studies in most marine habitats [1].

Marine caves are common features of rocky coastlines. Despite the lack of cave inventories for a large stretch of the Mediterranean coastline, more than 3000 marine caves have been reported in the region so far [22]. Because they host a biota rich in cave-exclusive, threatened and relict species, caves have been considered as "biodiversity reservoirs" for many animal groups [23,24]. Darkness and isolation mostly drive the composition and distribution of cave-dwelling communities [22,25,26]. These environmental parameters vary horizontally along the cave, establishing gradients that also depend on the specific topography of each cave [27,28]. Accordingly, marine caves significantly contribute to the overall heterogeneity of coastal areas, thereby favouring an increased species richness at regional scales. Moreover, caves provide numerous ecosystem services to humans, insofar as they act as habitat and nursery grounds for many commercially important species and serve as popular locations for marine recreational activities. However, the biodiversity of motile invertebrates within such environments and their role in the functioning of cave ecosystems remain largely unexplored. Indeed, most studies investigating motile macroinvertebrates in benthic habitats neglect marine caves. Simultaneously most studies focused on caves predominantly concentrate on conspicuous animals [1,4,29,30]. In the Mediterranean Sea, where marine caves have been historically better investigated than elsewhere in the world [22], recent reviews have summarized the diversity of cave sponges [23], fish [31] and decapods [32]. The synthesis conducted by Romano et al. [33] on benthic foraminifera within Mediterranean marine caves probably constitutes the only review specifically addressing small-sized inconspicuous biota, although it is primarily concerned with highlighting the reliability of this group as a paleoecological indicator rather than characterizing their biodiversity or ecological patterns on a regional scale.

Mediterranean cave-dwelling amphipods were first studied by Sandro Ruffo of the Verona National History Museum during the 1940s and 1950s. Ruffo worked on species from both marine and freshwater cave environments [34,35], see also references in [36]. Many of the sites sampled during that period were described as "marginal caves" or "mixohaline hypogean waters" [37,38], corresponding to the term "anchialine caves", which was introduced in the 1980s to characterize subterranean water bodies with an underground connection to the ocean [38,39]. Ruffo's investigations on anchialine habitats resulted in the description of novel taxa belonging to genera *Metahadzia* and *Salentinella* from Puglia (Southern Italy). Later on, exhaustive field and taxonomical studies were conducted by Karaman and Sket along the Dalmatian karstic coast (see [40] and references therein). Marine cave species, instead, were studied in the Tyrrhenian Sea and the French coast [35,37,41–43]. Studies on Mediterranean amphipod cave fauna continue to this date, but information on this topic is scattered among different sources such as taxonomic

descriptions of new species (e.g., [44]), species inventories (e.g., [45,46]) or ecological studies published in grey literature (e.g., [47]). In addition, most published research focused on the Western Mediterranean basin and the Adriatic Sea, while amphipod fauna in marine caves of the Eastern basin and the African coast remain understudied [48].

In the context of baseline assessments and ecological studies, it is imperative to pay more attention to the motile benthic macrofauna, since this group holds significant importance in comprehending the biodiversity and functioning of marine habitats. In an extensive review about Mediterranean marine caves, Gerovasileiou and Bianchi [22] reported on the occurrence of 83 amphipod taxa from 19 studies, but no detailed checklist was provided. Here, we summarize the existing knowledge about marine and brackish-water amphipods in Mediterranean marine and anchialine caves. First, we collected new information on the amphipod diversity of marine caves of the understudied Eastern Mediterranean basin through targeted samplings. New and previous data for other Mediterranean regions were combined in order to provide an updated inventory of the species reported from cave environments. Finally, we present the geographical and ecological patterns of amphipod fauna in Mediterranean marine and anchialine caves.

## 2. Materials and Methods

### 2.1. Amphipod Sampling

New records of amphipods were gathered between 2010 and 2022 from 13 marine caves, spanning from the East Ionian Sea (eight marine caves in Zakynthos Island) to the Aegean Sea (two caves in Lesvos Island, two in Crete and one in Rhodes) (Figure 1). In each cave, samples of dominant primary substrates (e.g., sponges, cnidarians, and bryozoans) were collected with SCUBA diving from different cave zones (a brief description of the depth, cave topography, etc., is provided in Supplementary Table S1). The primary substrates were first enclosed in plastic containers to prevent loss of epifauna, and then were carefully detached. In the caves of Lesvos Island, samples were collected using a quadrat sampler (20 × 20 cm) [49] at 10 stations (six in Fara cave and four in Agios Vasilios cave; with three replicate samples per station), located on the cave walls and ceilings along the horizontal cave axis (Figure 2—for detailed information see [50–52]). In the laboratory, all samples (already preserved in 70% ethanol) were washed through a 0.5 mm mesh sieve. Amphipods were identified to species level and counted.

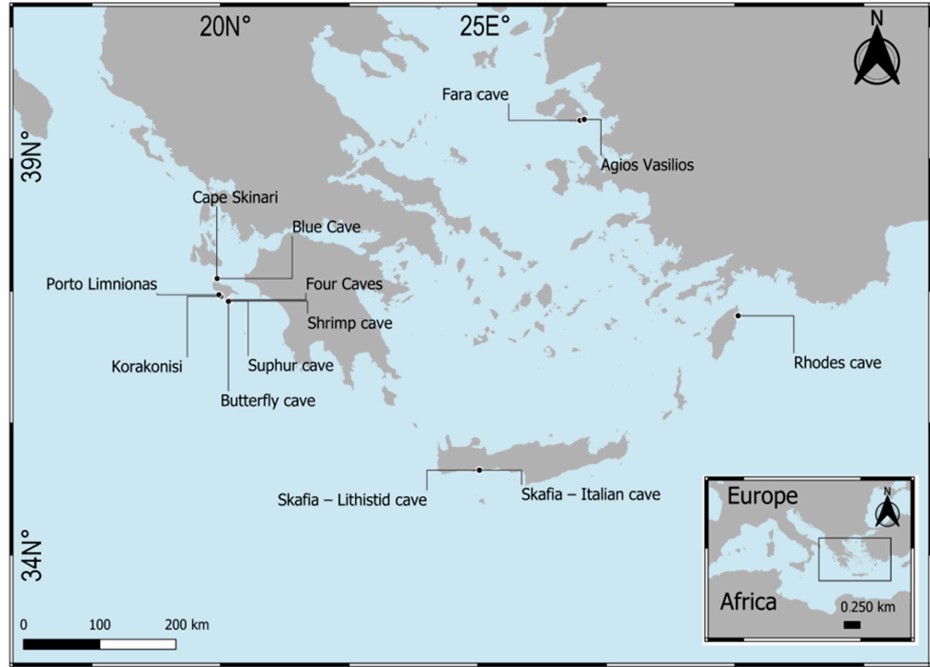

**Figure 1.** Location of the marine caves providing new material included in the present study.

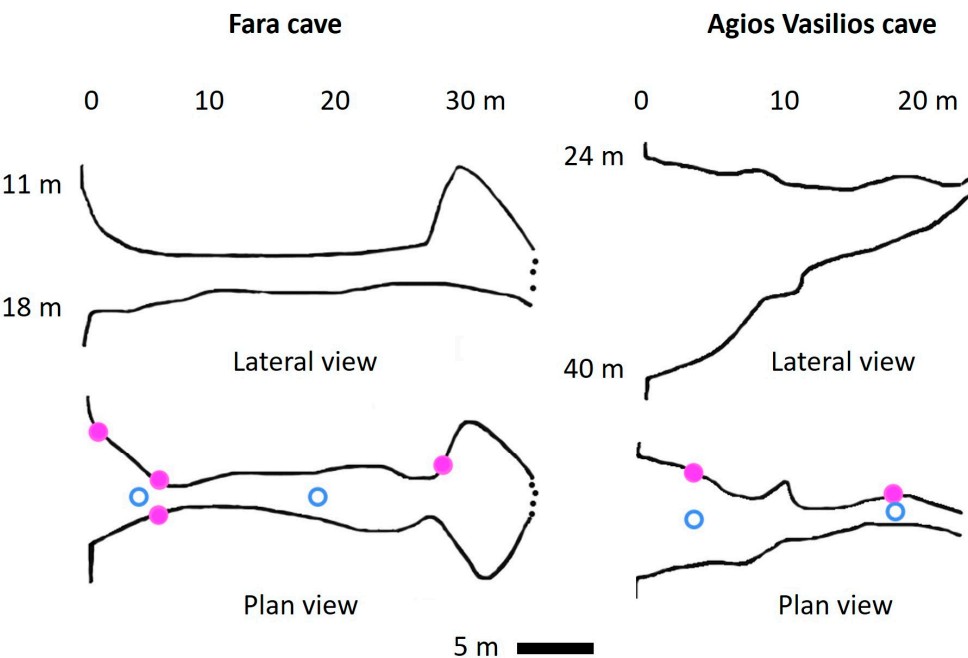

**Figure 2.** Lateral and plan views of Fara (**left**) and Agios Vasilios caves (**right**) of Lesvos Island. Location of collected samples is indicated in relation to distance from the entrances. Open blue circles: ceiling samples; full violet circles: wall samples. Modified from [51].

*2.2. Literature Review*

We reviewed the existing knowledge about marine and brackish-water amphipods in Mediterranean marine and anchialine caves. The initial dataset was sourced from the Mediterranean marine cave biodiversity database by [24] and Stygofauna Mundi [53], followed by a thorough literature review from a wide variety of sources, including peer-reviewed articles and grey literature sources (e.g., PhD theses and technical reports) (a full reference list is available in Supplementary Tables S1 and S2). Original genus and species names were updated to the World Register of Marine Species taxonomic backbone (WoRMS, last accessed on 28 July 2023; [54]). The final amphipod checklist will be uploaded to the World Register of marine Cave Species (WoRCS, http://www.marinespecies.org/worcs/, last accessed on 28 July 2023; [55]), a thematic species database of WoRMS which aims at creating a comprehensive taxonomic and ecological database of species from marine caves worldwide.

The resulting dataset consisted of occurrence data by cave. Separate datasets were compiled for marine and anchialine caves since they are characterized by different environmental constraints and harbour a distinctive fauna. Marine caves, which open directly on the coastline or on the seafloor, present a greater water exchange with the open sea, whereas anchialine caves often lack such entrances, resulting in a greater isolation. Consequently, anchialine caves typically exhibit distinct freshwater, brackish and marine water bodies, with varying residence times [56]. In this review, records from interstitial habitats, coastal wells or other freshwater hypogean habitats were not considered. Consequently, only those amphipods collected from marine or brackish waters within anchialine cave systems are considered. Nearby anchialine caves located in limestone terrains are often part of the same karstic systems. Therefore, multiple reports of the same species from different but presumably connected caves were considered as single-study sites. Marine caves show high heterogeneity with regard to their submersion level (i.e., fully or semi-submerged), morphology (i.e., blind-ended or tunnel-shaped), bathymetry, substrate type (hard or soft bottom) and ecological zonation (i.e., entrance, semi-dark and dark zones, and associated biocoenoses) [22]. When available, this type of information was also catalogued to explore potential ecological patterns (detailed information about cave features and references is provided in Supplementary Table S1.

*2.3. Ecological Characterization of the Species and Data Analysis*

Once the species occurrence dataset was compiled, the data were grouped in eight Mediterranean regions according to [22] to investigate regional diversity patterns, namely Alboran Sea, Algero-Provençal Basin, Tyrrhenian Sea, Tunisian Plateau, Adriatic Sea, Ionian Sea, Aegean Sea and Levantine Sea. Spearman's rank correlation coefficient was calculated using the Social Science Statistics website (https://www.socscistatistics.com/, accessed on 1 August 2023) to investigate the relationship of amphipod species number with the number of publications and number of caves studied in each region, as a non-parametric measure of statistical dependence.

In addition, all identified amphipod species were assigned to different ecological categories with regard to their zoogeographic and bathymetric distribution, substrate category and feeding habits in order to explore possible patterns. Precisely, the species were classified into four zoogeographic categories based on the classifications suggested by [57–59] and the World Amphipoda Database of WoRMS [54]: (i) Mediterranean endemics; (ii) Mediterranean–Atlantic; (iii) Mediterranean–Atlantic–Arctic; and (iv) Cosmopolitan species. The classification of bathymetric distribution included four zones according to [57], namely intertidal, infralittoral, circalittoral and bathyal. However, many species presented a wide bathymetric range comprising several of the aforementioned zones. Concerning substrate preference, soft bottom amphipods were categorized into those reported from coarse sands, fine/medium sands, muddy bottoms and indifferent (i.e., those present in a wide spectrum of sediment types). In the case of amphipods inhabiting hard bottoms, information was compiled about the primary substrate where they were reported (macroalgae, Porifera, Cnidaria, Bryozoa, Annelida, Mollusca, etc.), based on data obtained from different sources [29,57,60–73]. It should be noted that the above ecological characterization applies to the overall distribution of amphipod fauna across the Mediterranean Sea in all types of marine habitats, given the limited ecological knowledge about cave-dwelling amphipods.

Amphipods were assigned according to their feeding habits based on previous data from gut content analyses [20]. Four categories were considered, specifically, detritivores (species with more than 95% of detritus in their gut content), herbivores (>50% of algae), carnivores (>50% of prey) and omnivores (>5% of different trophic sources). In a few cases where data were not available for a given species, the trophic category was assigned based on data about other species of the same genus. New data from Eastern Mediterranean marine caves was mainly qualitative (occurrence data by cave or substrate type), except for the two marine caves of Lesvos Island, where amphipod abundance (N) and species richness (S) were also obtained. One-way permutational ANOVA (PERMANOVA) was used to examine the variability of the above measures across the stations of each cave (factor, station; fixed with six levels for Fara cave and four levels for Agios Vasilios cave). Statistical analysis was undertaken using the PRIMER-E v.6 software package [74].

## 3. Results

*3.1. Research Effort Overview*

Information on marine and brackish-water amphipods reported from Mediterranean marine and anchialine caves was sourced from 98 studies (33 about marine and 66 about anchialine caves; Supplementary Tables S1 and S2). These studies included information about 42 marine caves and 50 anchialine systems (including complex karstic systems with multiple nearby entrances). Marine caves have been studied in six countries, namely Spain (6 studies in 10 caves), France (5 studies in 6 caves), Italy (14 studies in 10 caves), Croatia (1 study in 1 cave), Malta (5 studies in 3 caves) and Greece (3 studies, including this work, in 13 caves), and six regions: the Adriatic Sea (1 cave), the Tunisian Plateau (2 caves), the Aegean Sea (5 caves), the Alboran Sea, the Algero-Provençal Basin and the Ionian Sea (9 caves in each). In contrast, anchialine caves have been investigated in six countries, namely Spain (31 studies in 20 sites, all located in the Balearic Islands), France (2 studies in 2 sites), Italy (21 studies in 6 sites), Croatia (20 studies in 18 sites), Montenegro (1 study in

1 site) and Greece (4 studies in 3 sites), comprising the Algero-Provençal Basin (41 studies in 25 sites), the Adriatic Sea (20 studies in 19 sites) and the Ionian Sea (18 studies in 6 sites).

### 3.2. New Data from Marine Caves of the Eastern Mediterranean Sea

Our survey provided 14 new records of amphipods from marine caves of Greece (only eight species were previously known from marine caves of Greece). Considering the new records obtained in the present study, a total of 16 species has been reported from the Aegean Sea and 13 from the East Ionian Sea. Agios Vasilios cave (11 species) and Shrimp cave (8 species) had, respectively, the highest number of species in these two regions.

Our quantitative survey in the caves of Lesvos Island yielded a total of 247 individuals (64 from Agios Vasilios cave and 183 from Fara cave) that belonged to 13 species (including two taxa identified at the genus level and an unidentified taxon) (Table 1). Abundance per scraped quadrat sampler (400 cm$^2$) ranged from zero to 16 individuals in Agios Vasilios cave and up to 45 individuals in Fara cave. Mean amphipod density in Agios Vasilios cave was 5.333 individuals $\pm$ 1.597 (SE, standard error), while that of Fara cave was double, with 10.167 individuals $\pm$ 2.949 (SE) in the scraped quadrates. The most abundant species was *Leptocheirus bispinosus* (48.2% of all individuals), followed by *Perrierella audouiniana* (20.2%), and *Liljeborgia dellavallei* (11.3%). The highest abundance and species richness were found on cave walls and ceilings of the entrance and semi-dark cave zones, dominated by sponges and scleractinians, whereas extremely low abundances were retrieved from the dark cave zone (Table 1). Abundance and species richness differed significantly among sampling stations in both the Fara and Agios Vasilios caves (Table 2).

**Table 1.** Amphipods recorded in the Fara and Agios Vasilios caves. The mean number of individuals is presented for each species along with the total number of individuals per sampling station. For each sampling station, the location inside the cave (L w, Left wall; R w, Right wall; C, Ceiling; and Entr, Entrance), distance from the entrance (m), biocoenosis (Cor, Coralligenous Biocoenosis; SD, Semidark Cave Biocoenosis; Trans, Transitional Zone; and Dark, Dark Cave Biocoenosis) and main sessile biota (Rh, Rhodophytes; Sc, Scleractinian corals; Sp, Sponges; Sr, Serpulid polychaetes; and Br, Bryozoans) are indicated.

| | Fara Cave | | | | | | Agios Vasilios Cave | | | |
|---|---|---|---|---|---|---|---|---|---|---|
| **Sample location** | L w | C | R w | L w | C | Walls | C | Walls | C | Walls |
| **Distance from entrance (m)** | Entr | 5 | 5-10 | 5-10 | 15-20 | 20-30 | 5-10 | 5-10 | 15-20 | 15-20 |
| **Biocoenosis** | Cor | SD | SD | SD | Trans | Dark | SD | SD | Dark | Dark |
| **Dominant encrusters** | Rh | Sc-Sp | Sc-Sp | Sp | Sr-Br | Sr-Sp-Br | Sc-Sp | Sp | Sc-Sp-Sr | Sp-Sr |
| **Species Sampling stations** | F1 | FC1 | F2 | F3 | FC2 | F4 | VC1 | V1 | VC2 | V2 |
| *Apherusa* sp. | | | | 0.3 | | | | | | |
| *Colomastix pusilla* | 0.3 | 0.3 | | 0.7 | | | 0.3 | 3.0 | | |
| *Gammarus subtypicus* | | | | | | | 0.7 | | | |
| *Iphimedia carinata* | | 0.3 | | | | | | | 0.3 | |
| *Iphimedia* sp. | | | | | | | | 0.3 | | |
| *Leptocheirus bispinosus* | | 10.0 | 21.7 | | 5.3 | | 2.3 | | 0.3 | |
| *Leptocheirus pectinatus* | | | | | | | 3.0 | 2.3 | 0.3 | 0.3 |
| *Leucothoe spinicarpa* | | | | | | | 1.3 | 0.3 | | |
| *Liljeborgia dellavallei* | 0.3 | 3.7 | 2.3 | | | | 2.7 | 0.3 | | |
| *Lysianassina longicornis* | | | | | | | 0.3 | | | |
| *Perrierella audouiniana* | 2.0 | 10.0 | 0.7 | 3.0 | | | | 1.0 | | |
| *Stenothoe antennariae* | | | | | | | 0.7 | | | |
| Amphipoda unid. | | | | | | | 1.0 | | | 0.3 |
| **Total number of individuals** | 8 | 73 | 74 | 12 | 16 | 0 | 37 | 22 | 3 | 2 |
| **Mean abundance** | 1 | 8 | 8.3 | 1.3 | 1.7 | 0 | 2.3 | 4.0 | 0.3 | 0.3 |
| **Species richness** | 3 | 5 | 3 | 3 | 1 | 0 | 6 | 9 | 2 | 3 |

In the studied Ionian caves, since the primary substrate type of most samples consisted of sponges (*Agelas oroides* and *Chondrosia reniformis*), sponge-associated species such as *Colomastix pusilla* were those with the highest frequency of occurrence (it was found in 55% of the sponge samples collected) and the highest abundance (26 individuals) (Table 3). Other common species were *Gitana sarsi* (15 individuals found in four caves) and *Leptocheirus pectinatus* (8 individuals collected in two caves). Other erect invertebrates such as bryozoans

and hydroids proved to be a suitable habitat for amphipods (16 amphipod individuals in a single bryozoan colony in the Korakonisi cave), but the scarcity of these substrates in most of the caves explored prevented a detailed exploration of their associated fauna.

**Table 2.** One-way PERMANOVA results for amphipod species richness and abundance across the sampling stations of Agios Vasilios and Fara caves.

| Cave | Source | df | Species Richness | | | Abundance | | |
| | | | MS | Pseudo-F | P (Perm) | MS | Pseudo-F | P (Perm) |
|---|---|---|---|---|---|---|---|---|
| Agios Vasilios | Station | 3 | 12.56 | 10.76 | 0.005 | 93.56 | 13.37 | 0.006 |
| | Res | 8 | 1.17 | | | 7.00 | | |
| Fara | Station | 5 | 2.89 | 5.78 | 0.015 | 379.17 | 5.95 | 0.010 |
| | Res | 12 | 0.50 | | | 63.72 | | |

**Table 3.** Abundance of each amphipod species collected along the new targeted biodiversity surveys in the Ionian Sea. CS = Cape Skinari; FC = Four caves; SC = Sulphur cave; BC = Butterfly cave; PL = Porto Limnionas; AO = *Agelas oroides*; Br = Bryozoans; CO = *Chondrosia reniformis*; Hy = Hydroids.

| | CS | Blue Cave | | FC | Shrimp Cave | | SC | BC | Korakonisi | | PL |
| Substrate | AO | AO | CO | AO | AO | Hy | AO | AO | AO | Br | AO |
|---|---|---|---|---|---|---|---|---|---|---|---|
| **Species** | | | | | | | | | | | |
| *Apocorophium acutum* | 1 | | | | | | | | | | 1 |
| *Apolochus picadurus* | | | | | | | | | | 1 | |
| *Colomastix pusilla* | | 1 | | 17 | 1 | | | 6 | | | 1 |
| *Coxischyrocerus inexpectatus* | | | | | | | | 1 | | | |
| *Gitana sarsi* | | | | | | | 5 | | | 9 | 1 |
| *Leptocheirus bispinosus* | | | | | 3 | | | 1 | | | |
| *Leptocheirus guttatus* | | | | | | 1 | | | | | |
| *Leptocheirus pectinatus* | | | | | | 1 | | | 2 | 5 | |
| *Leucothoe spinicarpa* | | | | | | | | 1 | | | |
| *Phtisica marina* | | | | | | 1 | 1 | | | | |
| *Pseudoprotella phasma* | | | 1 | | | 1 | | | | | |
| *Stenothoe monoculoides* | | | | | | | | | | 1 | |
| *Stenothoe* sp. | | | | | | 1 | | | | | |

### 3.3. Amphipod Diversity in Mediterranean Marine Caves

A total of 106 amphipod species belonging to 38 different families was reported from 42 Mediterranean marine caves in six countries (Table 4). Families with higher numbers of species were Ischyroceridae (7 species), as well as Maeridae, Phoxocephalidae and Stenothoidae (6 species each). Approximately half (49%) of the species are known from a single marine cave, whereas 80% were found in less than five locations (Supplementary Table S3). Highest species richness was recorded in Cueva de Cerro-Gordo in Spain (32 species), followed by Grotte de L'île Plane in France and Grotta del Mago in Ischia, Italy (26 species each) and Grotta di Bergeggi in Liguria, Italy (20 species).

The most widespread species were *Colomastix pusilla* (known from 16 caves), *Phtisica marina* (13), *Pseudoprotella phasma* (12) and *Leucothoe spinicarpa* (10), often found in high densities. Other species with a documented high abundance at cave entrances or semidark cave sectors were *Autonoe rubromaculatus*, *Coxischyrocerus inexpectatus*, *Harpinia crenulata*, *Harpinia pectinata*, *Lembos websteri*, and *Stenopleustes nodifera* [42,75,76]. A few species, such as *Aristias neglectus* and *Harpinia pectinata*, were reported as common or highly abundant in the inner dark zones of marine caves [76,77]. However, most studies lack information about abundance estimations and/or the ecological zone where the amphipod species were found. Overall, 41 amphipod species were reported from cave entrances, 80 from semi-dark cave sectors, and 27 inhabited the dark cave biocoenosis.

**Table 4.** Number of marine caves where amphipod species have been reported in each Mediterranean biogeographical region. * New records from marine caves of Greece. [†] This record most likely corresponds to *Elasmopus vachoni* Mateus & Mateus, 1966.

| Family | Species | Biogeographical Region | | | | | | |
|---|---|---|---|---|---|---|---|---|
| | | Alboran Sea | Algero-Provençal Basin | Tyrrhenian Sea | Tunisian Plateau | Adriatic Sea | Ionian Sea | Aegean Sea |
| Ampeliscidae | *Ampelisca rubella* A. Costa, 1864 | - | - | 1 | - | - | - | - |
| | *Ampelisca serraticaudata* Chevreux, 1888 | - | - | - | - | - | - | - |
| | *Ampelisca truncata* Bellan-Santini & Kaim-Malja, 1977 | - | - | - | 1 | - | - | - |
| | *Ampelisca typica* (Spence-Bate, 1856) | 1 | - | - | - | - | - | - |
| Amphilochidae | *Amphilochus manudens* Spence Bate, 1862 | - | 1 | - | - | - | - | - |
| | *Apolochus neapolitanus* (Della Valle, 1893) | - | - | 1 | - | - | - | - |
| | *Apolochus picadurus* (J.L. Barnard, 1962) * | - | - | - | - | - | 1 | - |
| | *Gitana* cf. *abyssicola* G.O. Sars, 1892 | 1 | - | - | - | - | - | - |
| | *Gitana sarsi* Boeck, 1871 * | - | 1 | - | - | - | 4 | - |
| Ampithoidae | *Ampithoe ramondi* Audouin, 1826 | - | 2 | 1 | - | - | - | - |
| | *Pleonexes helleri* (Karaman, 1975) | - | - | 1 | - | - | - | - |
| Aoridae | *Aora spinicornis* Afonso, 1976 | - | 1 | 1 | - | - | - | - |
| | *Autonoe rubromaculatus* (Ledoyer, 1973) | - | - | 1 | - | - | - | - |
| | *Lembos websteri* Spence Bate, 1857 | 4 | 3 | - | - | - | - | - |
| | *Microdeutopus algicola* Della Valle, 1893 | 1 | - | - | - | - | - | - |
| Aristiidae | *Aristias neglectus* Hansen, 1988 | - | 5 | - | - | - | - | - |
| | *Perrierella audouiniana* (Spence Bate, 1857) * | - | 1 | 1 | - | - | - | 2 |
| Atylidae | *Nototropis swammerdamei* (H. Milne-Edwards, 1830) | - | - | 1 | - | - | - | - |
| | *Nototropis vedlomensis* (Spence Bate & Westwood, 1862) | - | 1 | - | - | - | - | - |
| Bogidiellidae | *Marinobogidiella tyrrhenica* (Schiecke, 1979) | - | - | 1 | - | - | - | - |
| Calliopiidae | *Apherusa bispinosa* (Spence Bate, 1857) | - | 2 | 1 | - | - | - | - |
| Caprellidae | *Caprella hirsuta* Mayer, 1890 | - | - | 1 | - | - | - | - |
| | *Caprella liparotensis* Haller, 1879 | - | 1 | 1 | - | - | - | - |
| | *Liropus minimus* Mayer, 1890 | - | - | 1 | - | - | - | - |
| | *Phtisica marina* Slabber, 1796 | 3 | 5 | - | - | - | 3 | 2 |
| | *Pseudolirius kroyeri* (Haller, 1879) | 4 | - | - | - | - | - | - |
| | *Pseudoprotella phasma* (Montagu, 1804) * | 4 | 2 | 2 | - | - | 3 | 1 |
| Colomastigidae | *Colomastix pusilla* Grube, 1861 | - | 5 | 3 | - | - | 6 | 2 |
| Corophiidae | *Apocorophium acutum* (Chevreux, 1908) * | 1 | 1 | 1 | - | - | 1 | - |
| | *Leptocheirus bispinosus* Norman, 1908 | 1 | 1 | - | - | - | 2 | 2 |
| | *Leptocheirus guttatus* (Grube, 1864) * | - | - | - | - | - | 1 | - |
| | *Leptocheirus longimanus* Ledoyer, 1973 | 1 | - | - | - | - | - | - |
| | *Leptocheirus pectinatus* (Norman, 1869) * | 2 | 3 | - | - | - | 2 | 1 |
| | *Monocorophium sextonae* (Crawford, 1937) | - | 1 | - | - | - | - | - |
| Cressidae | *Cressa cristata* Myers, 1969 | - | - | - | 2 | - | - | - |
| | *Cressa mediterranea* Ruffo, 1979 | - | 2 | - | 1 | - | - | - |
| Cyproideidae | *Peltocoxa marioni* Catta, 1875 | - | - | 1 | - | - | - | - |
| Dexaminidae | *Dexamine spiniventris* (Costa, 1853) | 1 | 5 | 1 | - | - | - | - |

Table 4. *Cont.*

| Family | Species | Biogeographical Region | | | | | | |
|---|---|---|---|---|---|---|---|---|
| | | Alboran Sea | Algero-Provençal Basin | Tyrrhenian Sea | Tunisian Plateau | Adriatic Sea | Ionian Sea | Aegean Sea |
| | *Dexamine spinosa* (Montagu, 1813) | 1 | 3 | 1 | - | - | - | - |
| | *Tritaeta gibbosa* (Spence Bate, 1862) | 1 | - | 3 | - | - | - | - |
| Gammaridae | *Gammarus subtypicus* Stock, 1966 | - | - | - | - | - | - | 1 |
| Hyalidae | *Apohyale crassipes* (Heller, 1866) | - | 1 | - | - | - | - | - |
| | *Protohyale (Protohyale) schmidtii* (Heller, 1866) | - | 1 | 1 | - | - | - | - |
| Iphimediidae | *Iphimedia carinata* Heller, 1866 * | - | - | - | - | - | - | 2 |
| | *Iphimedia eblanae* Spence Bate, 1857 | - | 2 | - | - | - | - | - |
| | *Iphimedia minuta* G.O. Sars, 1883 | - | 4 | - | - | - | - | - |
| Ischyroceridae | *Centraloecetes dellavallei* (Stebbing, 1899) | - | 1 | - | - | - | - | - |
| | *Coxischyrocerus inexpectatus* (Ruffo, 1959) * | 3 | - | 1 | - | - | 1 | - |
| | *Ericthonius brasiliensis* (Dana, 1853) | - | 1 | 1 | - | - | - | - |
| | *Ericthonius punctatus* (Spence Bate, 1857) | 1 | - | - | - | - | - | - |
| | *Jassa marmorata* Holmes, 1905 | - | 1 | 1 | - | - | - | - |
| | *Jassa slatteryi* Conlan, 1990 | 1 | - | - | - | - | - | - |
| | *Microjassa cumbrensis* (Stebbing & Robertson, 1891) | 2 | - | - | - | - | - | - |
| | *Plumulojassa ocia* (Spence Bate, 1862) * | - | - | - | - | - | - | 1 |
| Kamakidae | *Cerapopsis longipes* Della-Valle, 1893 | 4 | - | - | - | - | - | - |
| Leucothoidae | *Leucothoe oboa* Karaman, 1971 | 1 | - | - | - | - | - | - |
| | *Leucothoe spinicarpa* (Abildgaard, 1789) | 1 | 5 | 1 | - | - | 1 | 2 |
| Liljeborgiidae | *Idunella nana* (Schiecke, 1973) | - | - | 1 | - | - | - | - |
| | *Liljeborgia dellavallei* Stebbing, 1906 | 1 | 3 | 2 | - | - | - | 1 |
| Lysianassidae | *Lysianassa caesarea* Ruffo, 1987 | - | - | - | - | - | - | 1 |
| | *Lysianassa costae* H. Milne Edwards, 1830 | - | 5 | - | - | - | - | - |
| | *Lysianassa pilicornis* Heller, 1866 | - | 2 | - | - | - | - | - |
| | *Lysianassina longicornis* (Lucas, 1846) * | - | 3 | - | - | 1 | - | 1 |
| Maeridae | *Elasmopus rapax* Costa, 1853 | - | 1 | - | - | - | - | - |
| | *Elasmopus pectenicrus* (Spence Bate, 1862) † | 1 | - | - | - | - | - | - |
| | *Elasmopus pocillimanus* (Spence Bate, 1862) | - | 1 | - | - | - | - | - |
| | *Maera grossimana* (Montagu, 1808) | - | - | 1 | - | - | - | - |
| | *Maeropsis revelata* (Krapp-Shcickel, Martì & Ruffo, 1996) | - | - | - | 1 | - | - | - |
| | *Othomaera othonis* (H. Milne Edwards, 1830) | - | 1 | - | - | - | - | - |
| | *Quadrimaera inaequipes* (A. Costa in Hope, 1851) | 1 | 4 | 1 | 1 | - | - | - |
| Melitidae | *Melita palmata* (Montagu, 1804) | - | 1 | - | - | - | - | - |
| Nuuanuidae | *Gammarella fucicola* (Leach, 1814) | 3 | 2 | - | - | - | - | - |
| Oedicerotidae | *Deflexilodes acutipes* (Ledoyer, 1893) | - | - | - | - | - | - | - |
| | *Deflexilodes griseus* (Della Valle, 1893) | 3 | 1 | - | - | - | - | - |
| | *Monoculodes packardi* Boeck, 1871 | 1 | - | - | - | - | - | - |
| | *Perioculodes longimanus* (Spence-Bate & Westwood, 1868) | 5 | - | - | - | - | - | - |
| | *Pontocrates arenarius* (Spence Bate, 1858) | 1 | - | - | - | - | - | - |
| | *Synchelidium* cf. *longidigitatum* Ruffo, 1947 | 1 | - | - | - | - | - | - |

**Table 4.** *Cont.*

| Family | Species | Biogeographical Region | | | | | | |
|--------|---------|------------|----------|----------|----------|----------|----------|----------|
| | | Alboran Sea | Algero-Provençal Basin | Tyrrhenian Sea | Tunisian Plateau | Adriatic Sea | Ionian Sea | Aegean Sea |
| Phliantidae | *Pereionotus testudo* (Montagu, 1808) | - | 1 | 1 | - | - | - | - |
| Photidae | *Gammaropsis crenulata* Krapp-Schickel & Myers, 1979 | - | 1 | - | - | - | - | - |
| | *Gammaropsis dentata* Chevreux, 1900 | - | - | - | - | 1 | - | - |
| | *Gammaropsis maculata* (Johnston, 1828) | 4 | 2 | - | - | - | - | - |
| Phoxocephalidae | *Harpinia ala* Karaman, 1987 | 2 | - | - | - | - | - | - |
| | *Harpinia antennaria* Meinert, 1890 | 2 | - | - | - | - | - | - |
| | *Harpinia crenulata* Boeck, 1871 | 4 | - | 1 | - | - | - | - |
| | *Harpinia pectinata* Sars, 1891 | 4 | - | - | - | - | - | - |
| | *Hippomedon massiliensis* Bellan-Santini, 1965 | 2 | - | - | - | - | - | - |
| | *Metaphoxus fultoni* (Scott, 1890) | 2 | - | - | - | - | - | - |
| | *Metaphoxus gruneri* Karaman, 1986 | - | - | - | 1 | - | - | - |
| Pleustidae | *Stenopleustes nodifer* (G.O. Sars, 1883) | - | 1 | - | - | - | - | - |
| Podoceridae | *Parunciola seurati* Chevreux, 1911 | - | 2 | - | - | - | - | - |
| | *Podocerus variegatus* Leach, 1814 * | - | - | - | - | - | - | 1 |
| Pontogeneiidae | *Eusiroides dellavallei* Chevreux, 1899 | 1 | 3 | 1 | - | - | - | - |
| Stenothoidae | *Stenothoe antennulariae* Della Valle, 1893 * | - | - | - | - | - | - | 1 |
| | *Stenothoe cavimana* Chevreux, 1908 | 1 | 1 | - | - | - | - | - |
| | *Stenothoe dollfusi* Chevreux, 1887 | 1 | 1 | - | - | - | - | - |
| | *Stenothoe monoculoides* (Montagu, 1813) * | - | - | 2 | - | - | 1 | - |
| | *Stenothoe pieropan* Krapp-Schickel, 1996 | - | - | 1 | - | - | - | - |
| | *Stenothoe tergestina* (Nebeski, 1881) | 1 | 4 | - | - | - | - | - |
| | *Stenothoe* sp. | - | - | - | - | - | 1 | 1 |
| Talitridae | *Macarorchestia remyi* (Schellenberg, 1950) | - | - | 1 | - | - | - | - |
| Tryphosidae | *Lepidepecreum crypticum* Ruffo & Schiecke, 1977 | - | - | 1 | - | - | - | - |
| | *Orchomene humilis* (Costa, 1853) | - | 1 | 1 | - | - | - | - |
| | *Tryphosella minima* (Chevreux, 1911) | - | 1 | - | - | - | - | - |
| Uristidae | *Tmetonyx nardonis* (Heller, 1867) | - | 1 | - | - | - | - | - |
| Urothoidae | *Urothoe elegans* Spence Bate, 1857 | 1 | - | - | - | - | - | - |

### 3.4. Regional, Zoogeographic and Ecological Patterns for Marine Cave Amphipods

The Algero-Provençal Basin was the Mediterranean region with the highest number of recorded species (52 species), followed by the Alboran Sea and the Tyrrhenian Sea (42 and 37 species, respectively), whereas the remaining regions harboured much fewer species (Aegean Sea: 16, Ionian Sea: 13, Tunisian Plateau: 6, and Adriatic Sea: 2 species) (Figure 3). The number of surveyed caves was found to be positively correlated with the number of amphipod species found in each region ($r_s$ = 0.778, $p$ = 0.039). A positive correlation, although non-significant, was also found between the number of studies and species by region ($r_s$ = 0.727, $p$ = 0.063). Concerning the zoogeographic characterization, most cave amphipods belonged to Mediterranean–Atlantic species (50%), followed by Mediterranean endemics (24.5%), cosmopolitan species (16%) and Mediterranean–Atlantic–Arctic (8.5%) (Figure 4, Supplementary Table S4).

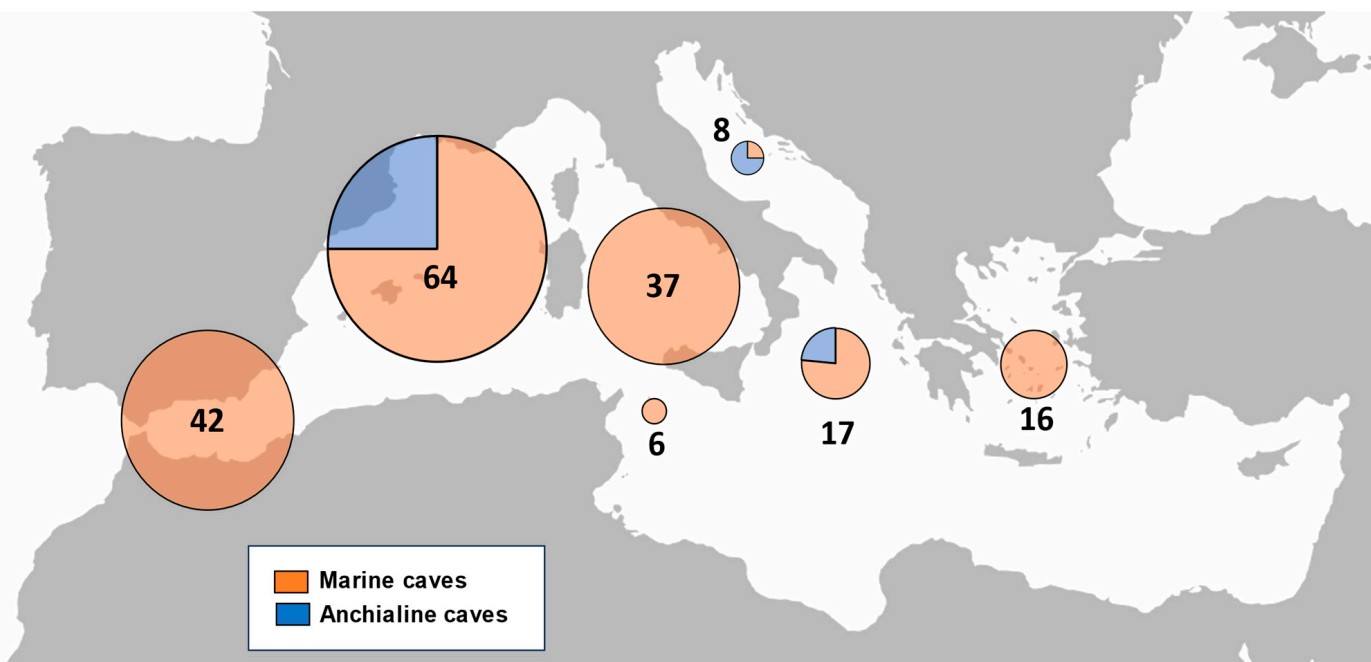

**Figure 3.** Number of cave-dwelling amphipod species reported in the Mediterranean. From left to right: Alboran Sea, Algero-Provençal Basin, Tyrrhenian Sea, Tunisian Plateau, Adriatic Sea, Ionian Sea and Aegean Sea. Percentage of species inhabiting marine caves (orange) and anchialine caves (blue) in each region is also provided.

Cave-dwelling amphipods inhabit a wide bathymetric range outside marine caves: 40% of the species range from shallow to circalittoral and/or abyssal depths (Figure 4), and only 13% of the species are exclusive to deep-sea bottoms (circalittoral, bathyal and abyssal species). Most species prefer hard substrates, including macroalgae, seagrasses and sessile invertebrates such as Cnidaria and Bryozoa (42 species each), Annelida (33 species), Porifera (25 species), and Mollusca (14 species). On the other hand, 30% of the species are known from soft bottoms, most of them with a wide ecological amplitude (40% of the soft-bottom species have been found in sediments with different granulometric composition). Finally, the main feeding strategy reported for cave-dwelling amphipods was detritivory (48%), followed by carnivory and omnivory (25% each), while only a small percentage (2%) corresponded to herbivores (Figure 4).

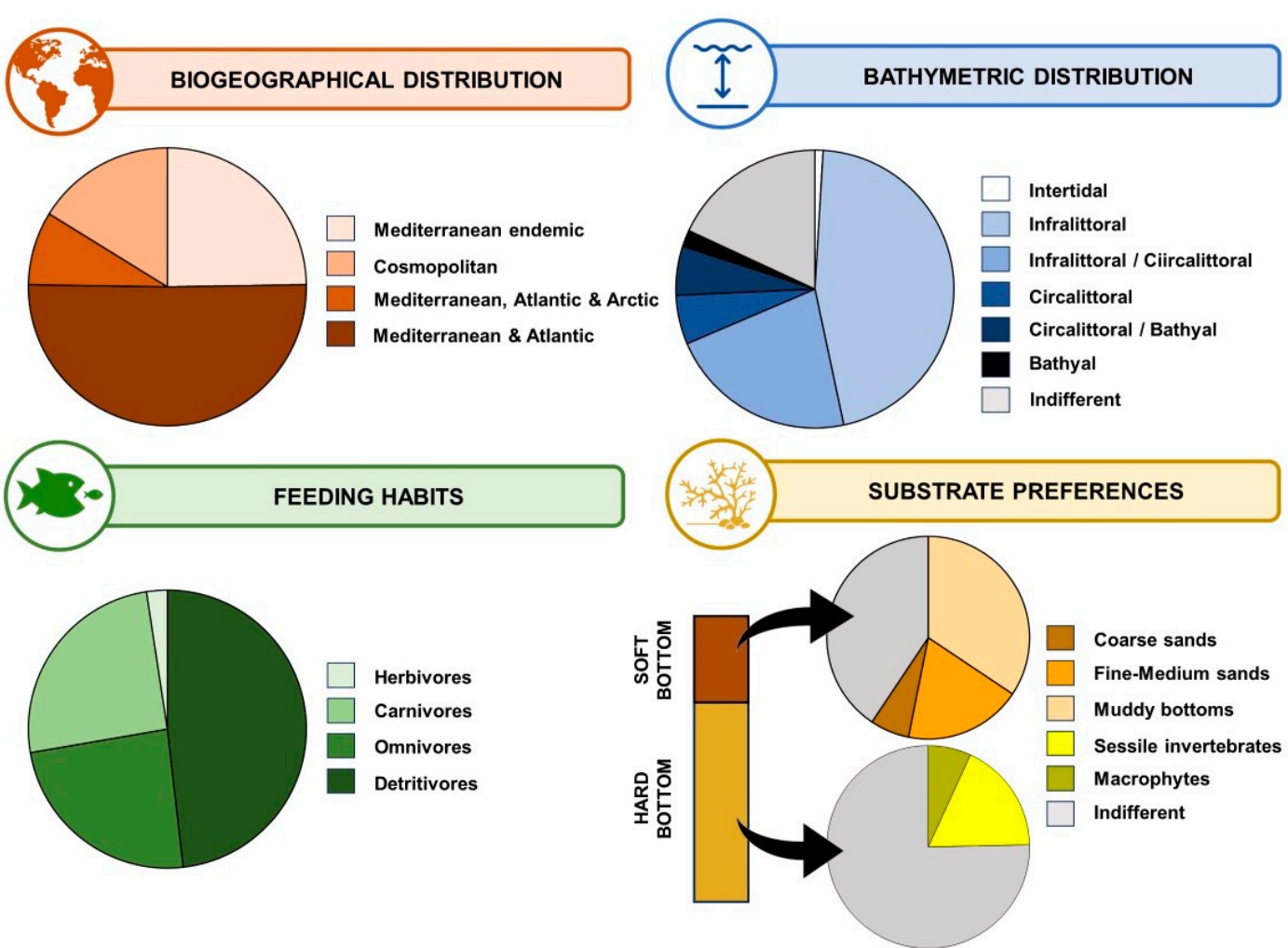

**Figure 4.** Ecological characterization of the amphipods inhabiting Mediterranean marine caves.

### 3.5. Amphipod Diversity in Mediterranean Anchialine Caves

Overall, 17 marine and brackish-water amphipod species have been reported from marine and brackish water layers of anchialine caves in the Mediterranean Sea (Table 5). Five of them belong to the family Niphargidae (genus *Niphargus*), and three to the family Bogidiellidae (genera *Bogidiella* and *Racovella*). Most species show a very narrow distribution, known only from a single cave or karstic system (Supplementary Table S2). The only exception was that of the species *Salentinella angelieri*, which is widespread across caves in the Algero-Provençal Basin, the Adriatic and Ionian regions. The Algero-Provençal Basin shows the highest number of species, with seven species from the Balearic Islands, three in the French coast and one species from Sardinia (Italy). Four amphipod species have been recorded from the Ionian Sea and six in the Adriatic Sea (Figure 2). In the latter, the species *Niphargus heberi* is known from brackish waters of 18 different anchialine caves along the Dinaric karst. The lack of data from other Mediterranean regions and information about the ecology of most species did not allow for the investigation of regional and ecological patterns.

**Table 5.** Number of anchialine caves where the amphipod species have been reported in each Mediterranean biogeographical region. Al-Pr Basin = Algero-Provençal Basin.

| Family | Species | Biogeographical Region | | |
|---|---|---|---|---|
| | | Al-Pr Basin | Adriatic Sea | Ionian Sea |
| Bogidiellidae | *Bogidiella balearica* Dancau, 1973 | 5 | - | - |
| | *Bogidiella cerberus* Bou & Ruffo, 1979 | - | - | 1 |
| Bogidiellidae | *Racovella birramea* Jaume, Grácia & Boxshall, 2007 | 1 | - | - |
| Cheluridae | *Chelura terebrans* Philippi, 1839 | 1 | - | - |
| Eriopisidae | *Psammogammarus burri* Jaume & García, 1992 | 1 | - | - |
| Hadziidae | *Hadzia fragilis* S. Karaman, 1932 | - | 6 | - |
| | *Metahadzia minuta* (Ruffo, 1947) | - | 2 | 3 |
| Metacrangonyctidae | *Metacrangonyx longipes* Chevreux, 1909 | 13 | - | - |
| Niphargidae | *Niphargus angelieri* Ruffo, 1954 | 1 | - | - |
| | *Niphargus delamarei* Ruffo, 1954 | 2 | - | - |
| | *Niphargus hebereri* Schellenberg, 1933 | - | 18 | - |
| | *Niphargus pectencoronatae* Sket & G. Karaman, 1990 | - | 3 | - |
| | *Niphargus salonitanus* S. Karaman, 195- | - | 1 | - |
| Pseudoniphargidae | *Pseudoniphargus leucatensis* Bréhier & Jaum, 2009 | 1 | - | - |
| | *Pseudoniphargus mercadali* Pretus, 1988 | 1 | - | - |
| Salentinellidae | *Salentinella angelieri* Delamare-Deboutteville & Ruffo, 1952 | 17 | 2 | 2 |
| | *Salentinella gracillima* Ruffo, 1947 | - | - | 3 |

## 4. Discussion

### 4.1. Marine Caves

Mediterranean marine caves host a considerable number of amphipod species, accounting for approximately 20% of the total Mediterranean amphipod fauna. Biodiversity assessments dating from the early 2000s reported from 249 to 466 amphipod species inhabiting the Mediterranean Sea [78,79]. However, considering the continuously increasing rate of amphipod species description in recent decades and the amount of research conducted in the area [8,80], along with the introduction of non-indigenous species [81], current estimates would be around 500 species. Nearly 40% of these species are endemic [78], while a slightly higher percentage (≈50%) of regional endemicity has been found on a global scale (i.e., half of the known amphipod species are exclusively distributed in a single bioregion) [7]. In contrast, around 25% of the amphipod fauna in Mediterranean marine caves are Mediterranean endemics. This relatively low percentage is in agreement with previous studies and the ecological patterns observed here (discussed below), suggesting that most amphipods in marine caves are generalists with wide ecological and geographical distribution [75]. Low endemism in marine caves has also been observed for other motile taxa, such as decapods (Mediterranean endemics cover only 7.9% of the cave-dwelling fauna) [32], while for sessile invertebrates such as sponges, Mediterranean endemics cover much higher percentages in marine caves (41.4%) [23]. In any case, there is a confused picture of the distribution of many species resulting from erroneous identifications, scarcity of information, existence of species complexes, etc. Some of the species previously reported from marine caves belong to unresolved species complexes or have a cryptogenic status at the Mediterranean (i.e., species that cannot be easily demonstrated as being either native or introduced) such as *Monocorophium sextonae*, *Ericthonius brasiliensis*, *Elasmopus rapax*, and *Elasmopus pectenicrus* (report of the latter is most likely a misidentification of *Elasmopus vachoni*) [82]. The only confirmed introduced species are *Jassa slatteryi* and *Jassa marmorata* [83,84], reported from marine caves in Spain (Alboran Sea) and Italy (Ligurian and Tyrrhenian Sea), respectively, [85] and this study.

Nine out of the ten dominant families (in terms of species number) in Mediterranean marine caves are also the most diverse amphipod families worldwide, so that a particular affinity to cave ecosystems cannot be inferred [7]. Those families (Ischyroceridae, Phoxocephalidae, Caprellidae, Maeridae, Stenothoidea, Oedicerotidae, Corophiidae, Ampeliscidae and Aoridae) often inhabit a wide variety of habitats such as intertidal rocky

shores, macroalgal forests, artificial environments (e.g., ports), coralligenous outcrops or even abyssal plains, to name a few [57]. Talitridae is the only highly diverse amphipod family scarcely represented in Mediterranean marine caves. This was expected since this family is usually associated with coastal intertidal or supralittoral habitats (sandhoppers) and mostly terrestrial habitats (landhoppers). The only talitrid reported in Mediterranean marine caves is *Macarorchestia remyi,* originally described from a freshwater littoral cave in Corsica [44] and later collected at about 200 metres from the entrance inside Blue Marino Cave (Sardinia), on *Posidonia oceanica* debris [86]. Despite these two first records in subterranean environments, *M. remyi* cannot be considered as a cave-dwelling species but as a driftwood hooper on sandy beaches [87,88]. Some eye-lacking talitrids are considered as truly troglobitic species, but all of them are terrestrial amphipods and none has been reported from Mediterranean shores [89,90].

None of the amphipod species reported in Mediterranean marine caves are exclusive to cave habitats, since all have been reported from other habitats as well [57]. This wide ecological amplitude is also supported by the high variety of substrates occupied by most species. Most of the hard bottom species reported here occur both on macrophytes and sessile invertebrates and more than 40% of the species inhabit soft bottoms with different granulometry (although, inside caves, they are usually found in muddy sediments, which prevail there due to the reduced hydrodynamism). Characteristic examples of amphipods commonly found in marine caves are the caprellids *Phtisica marina* and *Pseudoprotella phasma*, which are generalist species inhabiting macroalgae, seagrasses, sponges, cnidarians, bryozoans, ascidians and sediments of different granulometry [62,64,91,92]. Other abundant species in marine caves, such as *Leucothoe spinicarpa*, *Autonoe rubromaculatus*, *Lembos websteri* and *Stenothoe tergestina*, can also be found in different Mediterranean habitats such as macroalgal forests, seagrass meadows, coralligenous outcrops and maërl beds [93–95]. On the other hand, the commensal *Colomastix pusilla* is one of the few species common in marine caves that exhibit specialized habitat requirements, inhabiting the aquiferous system of different sponge species (e.g., *Agelas oroides*, and *Aplysina aerophoba*) [46,96]. *Colomastix pusilla* occurs also outside marine caves [95,97,98] but, considering that these environments are sponge biodiversity hotspots [23], marine caves may possibly play an important role in maintaining populations of sponge-associated invertebrates in the littoral zone.

The importance of sponges and sponge-dominated communities, as substrates sustaining amphipod assemblages in marine caves, was highlighted by the results of this work (high abundance and diversity values in Tables 1 and 3) and previous studies [46,96]. On the other hand, the absence of large-sized sponges (and other biotic substrates) in the dark cave sectors, along with the poor trophic resources, could explain the scarcity of amphipods in the dark cave biocoenosis. Some species commonly found in dark cave sectors (although generally at very low abundances) include widely distributed species found in various habitats such as *Aristias neglectus*, *Harpinia pectinata*, *Leucothoe spinicarpa*, *Leptocheirus bispinosus*, *Leptocheirus pectinatus*, and *Phtisica marina* [42,76,77,99] and this study. This finding of cryptobiotic (crevicular) and/or bathyphilic species (species with deep-water affinities), that originate from external marine environments, inside dark caves forms the basis of a concept known as "secondary stygobiosis" [22,100]. For instance, *Monoculodes packardi*, *Deflexilodes acutipes*, *Harpinia ala*, and *Parunciola seurati* are circalittoral or even bathyal species [57,101], whose shallowest records are known from marine caves. The species *Stenopleustes nodifer* (likewise other pleustid species) is typically associated with cold-water corals in deep-sea environments, but hundreds of individuals have been collected at 20 m depth, on the gorgonian *Paramuricea clavata* and on other sessile invertebrates in a French cave (Grotte de L'île Plane; [42]). Although most of the marine caves studied for their amphipod fauna are located at shallow depths (62% are shallower than 15 m deep and 88% do not exceed 20 m), more than half the amphipod species reported having deep-water affinities and approximately 13% are exclusive to the circalittoral or bathyal zone. Deep-water affinity of marine cave fauna has been associated with light deficiency, oligotrophic conditions and environmental stability shared by cave and deep-sea habitats

(e.g., [22,102,103]). These results provide further evidence about the wide ecological tolerance of the cave amphipod fauna, as well as the relevance and suitability of marine caves to improve our understanding of deep-water taxa.

Although marine caves are usually considered as isolated and "pristine" refuge habitats, many of the reported amphipod taxa are common inhabitants of heavily impacted areas such as harbours or fish farms. This is the case of *Apocorphium acutum*, *Apolochus* spp., *Cox-ischyrocerus inexpectatus*, *Ericthonius* spp. and the exotic species *Jassa slatteryi* [104,105]. On the other hand, other recorded amphipod species have been associated with unpolluted environments, such as *Harpinia pectinata*, *Microjassa cumbrensis* and *Stenothoe dollfusi* [106,107]. Amphipoda are considered as a good bioindicator group in a wide variety of habitats due to their high abundance and diversity, including species with different degrees of tolerance to variable natural and human stressors [21]. Considering the widespread distribution of many sensitive and tolerant amphipods inside marine caves, they could possibly be used as a suitable group to monitor environmental changes in these little-known ecosystems.

Concerning trophic habits, though the main feeding strategy of Amphipoda is detritivory [20], half of the cave-dwelling species seemed to feed on prey (being either strictly carnivorous or omnivorous). Marine caves are oligotrophic environments and thus cave dwellers may develop different strategies to cope with food scarcity, such as diurnal horizontal migrations, starvation resistance, chemo-litho-autotrophy, generalist diets and/or, conversely, niche specialization and resource partitioning [22,108–110]. Lack of herbivorous species and dominance of carnivores within amphipod cave fauna has been already highlighted in previous quantitative studies [28]. Trophic depletion is, in fact, the main factor accounting for the impoverishment of the amphipod community toward the inner section of the caves [76,99], as shown in the marine caves of the Aegean Sea studied here. The higher number of species recorded in semi-dark than in dark cave zones is also probably associated with the higher availability of trophic resources and suitable substrates, although this pattern could have been also affected by limited data availability [22].

Differences among Mediterranean regions in the number of amphipod species reported from marine caves should be interpreted with caution as these could be a result of skewed sampling effort. There is a clear bias towards the Northern Mediterranean, with no single amphipod record from marine caves of the North African and Middle East coasts. Along the northern Mediterranean coast, most studied caves, and consequently most of the amphipod species reported so far (as shown by the significant positive correlation), are located in the Western Mediterranean and specifically in parts of Spain, France and Italy. The marine cave with the highest number of species reported so far (Cerro Gordo cave, Spain) has been extensively sampled by specific studies addressing amphipod fauna [46,76,99]. In contrast, previous extensive checklists of amphipods inhabiting the Aegean and Levantine waters did not include records from marine caves [111–114]. Current knowledge on the amphipod assemblages inhabiting Mediterranean marine caves is far from complete, as shown from our targeted biodiversity surveys in marine caves of Greece, which have increased the number of cave-dwelling amphipods from the Eastern Mediterranean by 64%. In addition, our exhaustive review and new targeted surveys have increased the number of amphipod species reported from marine caves by 22% (from 83, according to [22], to 106 species), including six species reported for the first time as cave-dwellers, from Aegean (*Iphimedia carinata*, *Podocerus variegatus*, *Plumulojassa ocia*, and *Stenothoe antennulariae*) and Ionian caves (*Apolochus picadurus* and *Leptocheirus guttatus*).

Despite the small number of studies focusing on marine cave amphipods, their species number (106) resembles those reported from other marine habitats, such as *Posidonia* meadows (147 species), coralligenous beds (100 species) or deep bottoms (154 species) [94,101,115]. Therefore, as has been highlighted for other taxa [22,23,52,116] and ecological processes (i.e., nutrient cycling, resource storage) [117], cave ecosystems seem to play an extremely important role (in relation to their size) in amphipod biodiversity conservation. Cave exploration and biodiversity assessments on such habitats are still at their infancy. Thus,

descriptive faunistic studies conducted in any marine cave, and especially those addressing cryptic fauna, will probably provide novel and valuable information.

*4.2. Anchialine Cave Fauna*

Amphipod diversity patterns among anchialine caves in different Mediterranean regions are largely affected by the distribution of these fragmented habitats as well as sampling effort. To date, three areas within these regions stand out in terms of amphipod records: the Dinaric karst (Croatia and Montenegro), Puglia (Italy) and the Balearic Islands (Spain).

In contrast to the wide geographical and ecological distribution exhibited by marine cave amphipods, many of the species reported from marine and brackish waters in anchialine caves had limited distribution ranges. *Psammogammarus burri* was described from brackish waters of Cova des Burri (Cabrera, Balearic Islands) and has never been found elsewhere [118]. The species *Bogidiella cerberus* is known only from its type locality in Alepotrypa cave, in Peloponnese, Greece [119]. *Racovella birramea* is found exclusively in Cova des Coll (Mallorca Island). *Bogidiella balearica*, another Balearic endemic species, inhabits several anchialine caves in a small region of Mallorca and Cabrera Islands [120]. *Pseudoniphargus leucatensis* is only known from an anchialine cave on the French coast, which is also the only locality in our study where *Niphargus angelieri* is present. However, the latter has also been reported from other freshwater localities [121]. Despite being reported in a higher number of locations (>10 anchialine caves), *Metacrangonyx longipes* and *Niphargus hebereri* are also confined to a specific region. The former is exclusively distributed in Balearic groundwaters but, due to its tolerance to a wide range of salinity, it extends from freshwaters up to 200 m above the sea level to marine littoral caves [122]. *Niphargus hebereri* is endemic to the Dinaric karst, where it tolerates wide variations in salinity, oxygen concentration, and pollutants [123,124]. Moreover, *N. hebereri* and its congener *N. pectencoronatae* (also reported from brackish waters in anchialine caves) are listed as endangered in the Red List of Croatian Fauna due to their narrow distribution range and habitat specialization [125]. In any case, both *Niphargus* and *Pseudoniphargus* are highly diversified along the Mediterranean basin, but their distribution in anchialine caves is mainly restricted to freshwater areas [126–128]. Similarly, the family Hadziidae includes many subterranean species but only *Hadzia fragilis* and *Metahadzia minuta* are recorded in brackish waters of anchialine caves near the coast (e.g., [123,129,130]).

The only species with a wide distribution in anchialine caves is *Salentinella angelieri*, recorded in our census at the Algero-Provençal Basin, the Adriatic and the Ionian Sea. Some species of the genus *Rhipidogammarus* (e.g., *R. rhipidiophorus*) also display a wide circum-Mediterranean distribution associated with coastal groundwaters, including records from brackish waters in littoral caves with marine influence [131,132]. However, these taxa were not considered in the present study, since the aforementioned locations do not fall within the concept of anchialine caves. The family Salentinellidae currently includes 14 species of strictly subterranean aquatic amphipods but only *Salentinella angelieri* and *S. gracillima* are common inhabitants of brackish-water coastal aquifers and anchialine caves [133]. *Salentinella gracillima* is endemic to the coast of Puglia (Southern Italy), although its synonymy with *S. angelieri* has been recently suggested [133]. Overall, the distribution of *S. angelieri* encompasses the coasts of Spain, France, Italy, Croatia, Greece, Algeria, and Morocco [120,133–135]. Findings of this species in different subterranean biotopes and regions point out its high adaptability, so that it can be considered as a generalist species in coastal phreatic systems rather than as an anchialine specialist. In any case, *S. angelieri* may represent a species complex (Stoch, personal communication in [124]), which may be resolved by applying molecular and morphological approaches. Phylogenetic analyses proved to be valuable tools for revealing cryptic diversity and clarifying the phylogeography and evolutionary relationships of anchialine species [136]. Such tools have been already applied to some of the species herein reported, disclosing, for example, the

existence of fragmented population structures and independent lineages of *Pseudoniphargus mercadali* and *Metacrangonyx longipes* in different Balearic Islands [137,138].

Among the most common adaptations of anchialine amphipods to cave environments is the reduction of eyes and pigmentation, which could be observed in genera such as *Bogidiella*, *Racovella*, *Niphargus* and *Pseudoniphargus*. Other troglomorphic features associated with darkness include the elongation of body appendages and an increasing number of mechano- and chemoreceptors. The typically elongated third uropod of *Niphargus* and *Pseudoniphargus*, the long antennae and uropods of *Psammogammarus burri*, as well as the presence of peduncular protuberances in pleopods and uropods of *Bogidiella* and *Metahadzia* have also been interpreted in this context [120,139]. As they live in a buffered environment, cave species might exhibit lower tolerance to environmental changes than their surface counterparts. Some *Niphargus* species might be obligate stenothermic, particularly vulnerable to pollutants [123,140]. On the other hand, some environmental parameters may also fluctuate in anchialine caves affecting their amphipod fauna. As expected, anchialine amphipods such as *Niphargus hebereri*, *Salentinella angelieri* and *Metacrangonyx longipes* proved to be more tolerant to salinity changes than other species thriving on cave streams or at the boundary between the surface and subterranean ecosystems [141]. Oxygen availability may be also limited spatially or at periods (due to the lack of primary production) and some species (e.g., *N. hebereri*) develop behavioural and metabolic responses to such fluctuations [142]. Other life history and physiological adaptations to these extreme environments include reduced metabolic rate, increased longevity, delayed maturity, fewer eggs per clutch and infrequent reproduction [139,143,144]. However, these adaptations have not been thoroughly examined in anchialine amphipods, which constitute an interesting model group in order to study species diversification and test evolutionary and ecological hypotheses.

## 5. Conclusions

The present work highlights the high diversity of amphipods inhabiting Mediterranean marine caves. Our review showed that despite the relatively low number of studies dealing with amphipod fauna in marine caves, this habitat encompasses approximately 20% of the known Mediterranean amphipod species. The highest number of cave amphipod species has been found in the northwestern Mediterranean, which is the most studied area. Concerning the understudied Eastern Mediterranean basin, our 14 new records of cave amphipod species indicate the need for further exploration of invertebrate diversity inhabiting these unique ecosystems.

On the other hand, the ecological characterization of the cave amphipod fauna compiled herein unveiled the wide ecological amplitude of these assemblages. None of the 106 amphipod species reported in marine caves are exclusive of cave habitats, being also present in other marine habitats such as macroalgal forests, seagrass meadows or coralligenous outcrops. Most of the species may inhabit different substratum types and exhibit a generalist diet and wide bathymetric and geographical range.

In contrast to marine cave amphipods, species reported from Mediterranean anchialine caves often had narrow distribution range and showed habitat specialization. Many of the 17 amphipod species reported from marine-brackish waters in anchialine caves have been found exclusively in one or few cave localities and display morphological and physiological adaptations to the cave environment.

These biological and ecological patterns provide valuable insights that may contribute to the conservation of this highly diverse and yet overlooked biotic component of marine caves. Furthermore, the species inventory presented can serve as a baseline study to further increase our knowledge of Mediterranean cave amphipods.

**Supplementary Materials:** The following supporting information can be downloaded at https://www.mdpi.com/article/10.3390/d15121180/s1. Table S1: Marine cave description and references; Table S2: Amphipods in anchialine caves and references; Table S3: Amphipods in marine caves; Table S4: Ecological characterization of marine amphipods.

**Author Contributions:** Conceptualization, C.N.-B. and V.G.; methodology, C.N.-B., A.M., J.S.-V., S.C., M.D. and V.G.; sampling: C.N.-B., J.S.-V., S.C., M.D. and V.G.; identified samples, C.N.-B., M.D., W.P., E.V. and V.G.; formal analysis, C.N.-B., A.M., J.S.-V., S.C., M.D., W.P., E.V. and V.G.; writing—original draft preparation, C.N.-B.; writing—review and editing, A.M., J.S.-V., S.C., M.D., V.G., W.P. and E.V.; funding acquisition, C.N.-B. and V.G. All authors have read and agreed to the published version of the manuscript.

**Funding:** The present article benefited from two mobility grants awarded by the University of Seville and Spanish Ministry of Science and Innovation to C.N.-B. and J.S.-V., respectively. Additional funding was also provided by the project US-1381059 (Junta de Andalucía Proyectos I + D + I en el marco del Programa Operativo FEDER Andalucía) and TD2021-129725A (Spanish Ministry of Science and Innovation—Proyectos de Transición Ecológica y Transición Digital). S.C. was financially supported by doctoral fellowships by Agência Regional para o Desenvolvimento da Investigação, Tecnologia e Inovação (ARDITI-M1420-09-5369-FSE-000002). The material collection in Aegean marine caves was supported by (a) the Research Funding Programme "Heracleitus II: Investing in knowledge society" (EU Social Fund and Greek national funds); and (b) the project "Centre for the study and sustainable exploitation of Marine Biological Resources (CMBR)" (MIS 5002670), implemented under the Action "Reinforcement of the Research and Innovation Infrastructure", funded by the Operational Programme "Competitiveness, Entrepreneurship and Innovation" (NSRF 2014–2020) and co-financed by Greece and the EU (European Regional Development Fund).

**Data Availability Statement:** Data supporting all the reported results are publicly available at the Supplementary Materials here provided.

**Acknowledgments:** We are grateful to the Management Unit of Zakynthos and Ainos National Parks and Protected Areas of the Ionian islands of the Natural Environment and Climate Change Agency (NECCA) for providing sampling permission in marine caves of the Natura sites GR 2210001 and GR 2210002. We would also like to thank Dennis Mohr and the staff of Nero Sport Diving Center (Limni Keriou, Zakynthos), Charalampos Dimitriadis and Maria Sini for their valuable knowledge and help during sampling in Zakynthos and Lesvos. We would also like to thank Margaret Eleftheriou for reviewing the English language of the text and three anonymous referees for their helpful comments and suggestions.

**Conflicts of Interest:** The authors declare no conflict of interest. Funders had no role in the design of the study; in the collection, analyses, or interpretation of data; in the writing of the manuscript; or in the decision to publish the results.

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
