# Peer review of "Amphipods in Mediterranean Marine and Anchialine Caves: New Data and Overview of Existing Knowledge"

_diversity, doi:10.3390/d15121180_

Round 1

Reviewer 1 Report

Comments and Suggestions for Authors

This project was well-devised and has produced a well-written manuscript. It was interesting and informative for an amphipod researcher such as I, and I am hopeful that others in the invertebrate community will also read it with interest. Well done. Your results are thoughtful and include discussion of the hidden downfalls to working with literature-based citings and identifications of amphipod species. 

Author Response

We appreciate the positive feedback of the reviewer.  On behalf of the coauthors, we would like to acknowledge the time invested by the reviewer. 

Reviewer 2 Report

Comments and Suggestions for Authors

this was a very nicely written, useful and interesting review and i cammend you for this.

The only tiny errors i found were that sometimes you break the rule of giving numbers 10 and over as numerals, not words ( see lines 242, 381 for example)

Also do not capitalise words ( other then proper names of course ) in the titles of journal papers in reference list - see for example refs 32 and 33 (lines 730-732)

Author Response

Reviewer 2 (R2): This was a very nicely written, useful and interesting review and i cammend you for this. The only tiny errors i found were that sometimes you break the rule of giving numbers 10 and over as numerals, not words ( see lines 242, 381 for example)

(A): Changes have been conducted accordingly in lines 29, 125, 242, 381.

(R2): Also do not capitalise words (other then proper names of course ) in the titles of journal papers in reference list - see for example refs 32 and 33 (lines 730-732)

(A): A through correction of the reference list has been conducted in the revised version of the manuscript to fit journal guidelines and the comments suggested by reviewer 2 and 3.

On behalf of the coauthors, we would like to acknowledge the time invested by the reviewer. 

Reviewer 3 Report

Comments and Suggestions for Authors

The manuscript provides a comprehensive overview of the amphipods inhabiting Mediterranean marine and anchialine caves, a commendable effort that greatly contributes to our understanding of cave-dwelling amphipods. However, there are a few minor suggestions for improving the MS.

Firstly, the manuscript lacks a conclusion section. It would be beneficial to include a concise summary of the main findings and their implications for enhancing the knowledge of cave-dwelling amphipods.

Secondly, in the abstract, it is mentioned that 14 new record species have been identified. To enhance readability, it would be helpful to highlight these newly recorded species in Table 1.

Lastly, it is advised to carefully review the formatting of the references to ensure their compliance with the diversity requirements.

Comments on the Quality of English Language

The quality of English is OK, but it is best to find a native speaker to revise.

Author Response

We appreciate the possitive feedback and, on behalf of the coauthors, we would like to acknowledge the time invested by the reviewer

Reviewer 3 (R3): The manuscript provides a comprehensive overview of the amphipods inhabiting Mediterranean marine and anchialine caves, a commendable effort that greatly contributes to our understanding of cave-dwelling amphipods. However, there are a few minor suggestions for improving the MS. Firstly, the manuscript lacks a conclusion section. It would be beneficial to include a concise summary of the main findings and their implications for enhancing the knowledge of cave-dwelling amphipods.

(A): The following paragraph has been included at the end of the manuscript, summarizing our main results and the relevance of the patterns highlighted.

The present work highlights the high diversity of amphipods inhabiting Mediterranean marine caves. Our review showed that despite the relatively low number of studies dealing with amphipod fauna in marine caves, this habitat encompasses approximately 20% of the known Mediterranean amphipod species. The highest number of cave amphipod species has been found in the northwestern Mediterranean, which is the most studied area. Concerning the understudied Eastern Mediterranean basin, our 14 new records of cave amphipod species indicate the need for further exploration of invertebrate diversity inhabiting these unique ecosystems.

On the other hand, the ecological characterization of the cave amphipod fauna compiled herein unveiled the wide ecological amplitude of these assemblages. None of the 106 amphipod species reported in marine caves are exclusive of cave habitats, being also present in other marine habitats such as macroalgal forests, seagrass meadows or coralligenous outcrops. Most of the species may inhabit different substratum types and exhibit a generalist diet and wide bathymetric and geographical range.

In contrast to marine cave amphipods, species reported from Mediterranean anchialine caves often had narrow distribution range and showed habitat specialization. Many of the 17 amphipod species reported from marine-brackish waters in anchialine caves have been found exclusively in one or few cave localities and display morphological and physiological adaptations to the cave environment.

These biological and ecological patterns provide valuable insights that may contribute to the conservation of this highly diverse and yet overlooked biotic component of marine caves. Furthermore, the species inventory presented can serve as a baseline study to further enhance our knowledge of Mediterranean cave amphipods.   

(R3): Secondly, in the abstract, it is mentioned that 14 new record species have been identified. To enhance readability, it would be helpful to highlight these newly recorded species in Table 1.

(A): Changes have been conducted accordingly in Table 1

(R3): Lastly, it is advised to carefully review the formatting of the references to ensure their compliance with the diversity requirements.

(A): A through correction of the reference list has been conducted in the revised version of the manuscript to fit journal guidelines and the comments suggested by reviewer 2 and 3.

(R3): The quality of English is OK, but it is best to find a native speaker to revise.

(A): The manuscript has been already reviewed by Dr. Margaret Eleftheriou, who is a native speaker and also a professional English proof-reading editor (https://www.researchgate.net/profile/Margaret-Eleftheriou).